# The Impact of Divestment Announcements on the Share Price of Fossil Fuel Stocks

**Truzaar Dordi * and Olaf Weber** 

School of Environment, Enterprise and Development (SEED), University of Waterloo,
Waterloo, ON N2L 3G1, Canada; oweber@uwaterloo.ca
* Correspondence: tdordi@uwaterloo.ca; Tel.: +519-888-4567 (ext. 30194)

**Abstract:** Several prominent institutional investors concerned about climate change have announced their intention or have divested from fossil fuel shares, to limit their exposure to the industry. The act of fossil fuel divestment may directly depress share prices or stigmatize the industry's reputation, resulting in lower share value. While there has been considerable research conducted on the performance of the fossil fuel industry, there is not yet any empirical evidence that divestment announcements influence share prices. Adopting an event study methodology, this study measures abnormal deviations in stock prices of the top 200 global oil, gas, and coal companies by proven reserves, on days of prominent divestment announcements. Events are analyzed independently and in aggregate. The results make several notable contributions. While many events experienced short-term negative abnormal returns around the event day, the effects of events were more pronounced over longer event windows following the New York Climate March, suggesting a shift in investor perception. The results also find that divestment announcements related to campaigns, pledges, and endorsements all have a significant effect over the short-term event window. Finally, the results control for the general underperformance of the industry over the estimation window, attesting that the price change is caused by divestment announcements. Several robustness tests using alternate expected returns models and statistical tests were conducted to ensure the accuracy of the result. Overall, this study finds that divestment announcements decrease the share price of the fossil fuel companies, and thus, we conclude that 'divestors' can influence the share price of their target companies. Theoretically, the result adds new knowledge regarding the efficacy of the efficient market hypothesis in relation to divestment.

**Keywords:** divestment; fossil fuel; event study; efficient market hypothesis

## 1. Introduction

Increased concentrations of carbon dioxide emissions, affiliated with fossil fuel use [1], continue to accumulate in the atmosphere, raising global temperatures [2]. To prevent further global warming, carbon production must remain within a certain budget [3,4]. Consequently, most current fossil fuel reserves may not be burned [5].

To address the need to restrict increased greenhouse gas emissions, some institutional investors have put pressure on the fossil fuel industry to decrease their share of carbon emissions in the atmosphere [6–8]. Some shareholders have attempted to engage with fossil fuel companies by pressuring them to disclose information or act on climate; however, these efforts have resulted only in a few small successes [9]. Alternatively, a number of investors have announced to divest their shares in the fossil fuel industry for both financial [10] and ethical rationales [11].

Recent studies have analyzed the consequences of fossil fuel divestment. The results demonstrated that divestment did not result in lower risk-adjusted returns for the investors [12,13]. Hunt and

Weber [14], however, found positive effects of different levels of fossil fuel divestment on risk-adjusted returns of the Canadian Toronto Stock Exchange (TSX) 260 index. Finally, based on theoretical assumptions and evidence from other divestment campaigns, Ansar, Caldecott, and Tilbury [15] suggest small direct and indirect effects of divestment on the share price of the fossil fuel industry. Empirical studies on the effect of divestment announcements on the share price of fossil-fuel shares, however, do not exist yet.

Therefore, the purpose of this paper is to provide evidence on whether investors value divestment events, by examining the effect of divestment announcements on the stock price of the affected companies. This evidence is important theoretically and in practice, to understand whether investors are responding to recent calls for divestment and which mechanisms are at play in driving this response.

Consequently, we present an event study examining the effect of divestment announcements on the share-price of fossil-fuel industry members. In line with the efficient market hypothesis of modern portfolio theory, deviations in share prices around the event day would suggest that markets respond to divestment-related events. Twenty-four independent divestment announcements were examined for abnormal returns across the 200 fossil fuel companies with regard to carbon emissions content of their reported reserves. Deviations in share prices are benchmarked against the MSCI Inc. all-country world index (ACWI) and tested for significance using the standardized cross-sectional Boehmer, Musumeci and Poulsen (BMP) test. The results suggest a significant negative impact of divestment announcements on the share price of the fossil fuel companies in our sample.

This study contributes to divestment theory and practice by broadening knowledge on the effects of divestment announcements on the share price of firms. It is the first of its kind to examine the effect of an ethical investor-driven event on share prices of a group of companies. This study is in line with the price pressure hypothesis, that investors who accommodate demand shifts, for instance, through divestment, are compensated [16] regardless of delivering new market information. However, we added knowledge about the validity of the price pressure hypotheses in an ethical investment context. Second, this study contributes to the event study literature adding knowledge about the impacts of events on industries, instead of individual shares. Finally, we demonstrate that ethically driven investment announcements by a powerful and legitimate stakeholder that address an urgent problem, such as climate change, have impacts on the share price of fossil fuel stocks. We conclude that financial markets price divestment announcements as market risks.

This paper is organized as follows. Section two provides an in-depth look at the factors driving the divestment campaign, the burgeoning body of literature using the event study methodology to examine market response, and on the theoretical framing of key mechanisms influencing the impact of divestment. Section three discusses the data and methodology. Section four highlights the empirical results. Finally, section five concludes the analysis with discussion.

## 2. Literature

This paper bridges several strands of literature. First, we discuss the divestment literature. Secondly, the literature on event studies will be presented. Thirdly, we will introduce the theories on which this study is based.

### 2.1. Divestment

Divestment is defined as a socially motivated decision by private wealth owners or institutional investors to withhold capital from firms involved in perceivably reprehensible activities [15,17]. Divestment as a social movement has been adopted in the past, most recognizably against human rights violations of the Apartheid system, as well as against the tobacco industry [18,19], war crimes [20], and other 'sin stocks' [21,22].

Socially-responsible investors use divestment to pressure fossil fuel companies to change. Pressure is achieved through direct effects of divestment on their stock prices or through the reputational damage that divestment or divestment announcements can make [15]. There are two main motivations

for investors to divest. First, investors may choose to divest to purposefully depress the share prices of fossil fuel stocks and pressure the companies to change their practices [23]. Secondly, they might divest because they expect reduced financial returns of fossil fuel-related investments caused by stranded assets through the rapid devaluation of fossil fuel reserves [14,15,24].

Recent studies have analyzed the financial returns of fossil fuel investments [12,13,15,25], suggesting that divestment does not impair portfolio performance. Trinks et al. [13] suggest that fossil fuel companies have not contributed to portfolio diversification since as early as 1927. Hunt and Weber [14] find that divestment not only results in higher risk-adjusted returns but also reduces the carbon exposure of investment portfolios. Therefore, some investors try to divest from their fossil-fuel holdings to decrease the GHG exposure of their portfolio [26].

Conversely, divestment may establish an anti-fossil fuel norm [27] intended to stigmatize the industry and delegitimize the industry's political, economic, and social license to operate [28]. Proponents of divestment argue that institutional investors have a moral obligation to divest [29] as financial gains should not be achieved through investments that harm the climate. Notably, while divestment communicates a symbolic message, concerns have been raised around environmental justice related to decision-making power, rights, and diversity [30]. Moreover, institutional divestors may be more motivated by market-driven solutions than proactively mitigating the financial risks associated with carbon emissions and climate change [31].

Finally, divestment can influence the fossil-fuel industry [32] by raising the costs of capital [33], restricting corporate solvency to explore and exploit new reserves, and weakening production capacity in the long run [34]. Notably, even if there is no impact of divestment announcements on the stock price, the reputational damage could be impactful enough to achieve change in the industry, as literature on corporate reputation shows [35–37].

## 2.2. Event Studies

Event studies assess the impact of an event on a firm or industry. They analyze the events' effects over a relatively short period through shifts in stock prices rather than over a longer period through productivity-related outcomes [38]. Event studies test the efficient market hypothesis, claiming that stock prices adjust to reflect all newly available information that is relevant for the value of a firm [39,40].

There is a vast body of literature on event study methodology [41–46]. This literature emphasizes the pertinence and robustness of the event study methodology and offers a valuable background for studying the impact of the fossil fuel divestment movement.

The literature that applies the event study methodology is manifold [38,47]. Kothari and Warner [48] estimate that over 500 published studies have adopted the event study method to measure the impact of corporate governance strategies, mergers and acquisitions, financing decisions, post earnings announcements, chief executive successions, effects of regulation, and changes in taxation [49].

Several publications apply event studies to measure events related to corporate social responsibility. Cheung [50] found a significant but temporary positive (negative) impact of inclusion (exclusion) from the Dow Jones Sustainability World Index on the share prices or the respective firms. Amer [51] finds a similar result for member organizations that fail to report to the United Nations Global Compact (UNGC). Ammann et al. [52] found a significant positive impact of a Morningstar Sustainability Rating on mutual fund flows persisting up to one year after the announcement.

Studies analyzing the impact of events and announcements conducted by others besides the company itself also found significant impacts on share prices. These results correspond with the price pressure hypothesis stating that investors who accommodate demand shifts have to be compensated [16]. A study by Klassen and McLaughlin [53], for instance, found that positive news about a firm's environmental performance, such as a performance award announcement, increased the share price of the company while negative announcements, such as environmental fines, decreased the value of the share price. A study analyzing credit announcements for high carbon emitters found positive impacts

for low carbon emitters and negative impacts on the share price of high carbon emitters [54]. Rubin and Jones [37], however, only found effects of negative announcements about environmental events if they harm customers or suppliers of firms, or if they harm a firm's reputation. Finally, similar works on the fossil fuel industry measured the impact of announcements related to the 21st Conference of Parties (COP21) on a variety of fossil fuel shares and found moderate effects [55].

Six event study publications examined the impact of the Anti-Apartheid divestment movement on the performance of foreign companies and banks involved in Apartheid South Africa [56–60]. Nuances in the design framework of these studies have led to conflicting results [61]. There is evidence that the fossil fuel divestment movement shares similarities with the anti-Apartheid divestment movement [25] and thus, this method of analysis is applicable in the context of fossil-fuel divestment. However, a common feature of these studies was that the companies themselves announced the divestment from South-African businesses.

To summarize, the listed studies found significant impacts of environmentally related events and announcements on share prices. There is no quantitative evidence, however, to support or reject the notion that fossil fuel divestment events are influential. Thus, the objective of this study is to understand whether divestment announcements have an impact on the share price of fossil fuel stocks.

As indicated above, the mechanisms and the heterogeneity of investors' motivations are complex, and all mechanisms must be acknowledged. Negative effects could arise from socially-responsible investors deciding to divest, but also from traditional investors concerned about portfolio performance due to the reputational risks associated with the stock they hold. While acknowledging the heterogeneity in motivations that could lead to a measurable impact, our event study attempts to capture the effect associated with divestment announcements at the market level.

## 2.3. Efficient Market Hypothesis

Theoretically, the efficient market hypothesis [62], a subset of modern portfolio theory, delivers one possible explanation for whether fossil fuel divestment events influence the share prices of fossil fuel companies. The theory proposes that markets are efficient at reflecting and compounding all available information in the price of a security [63]. The efficient market hypothesis is tested through event studies to examine the process by which share prices respond to certain kinds of new information [40]. Seminal works by Heinkel, Kraus, and Zechner [64] and Fama and French [65] suggest that divestment of fossil fuel company shares is unlikely to have any direct share price effects, as at least twenty percent of total assets under management would have to divest collectively. Moreover, divestment announcements represent a commitment of the funds to divest over time, further attesting that divestment events might not impact the fossil fuel shares directly. If, however, divestment announcements do have a measurable impact on fossil fuel shares, we consider other mechanisms at play, such as stakeholder influence on reputation.

## 2.4. Stakeholder Theory and Social Norms

In combination with the efficient market hypothesis, stakeholder theory [66] may explain the impact of fossil fuel divestment announcements by institutional investors. While stakeholder-related performance is financially relevant but priced-in [67], unanticipated events may have direct effects on share prices to reflect the impact of the information. Stakeholders can 'explain and guide the structure and operations of the established corporation' [68], in much the same way shareholders can influence corporate objectives of a firm. Consequently, stakeholder pressure can also impact the share price of a firm through the act of announcing a divestment or the reputational impact it brings. A drop in the share price can be expected if the announcing stakeholder is powerful, for instance, as a powerful investor, owns legitimacy, or if there is an urgency, for instance, climate change [69].

There is also evidence that social movements by secondary (non-influential) stakeholders can illicit institutional change by actively influencing the conditions that led to dissatisfaction [70]. In the context of fossil fuel divestment, demonstrations of protests can be a relevant source of new information that

influences share prices [71]. While divestment may not directly threaten the demand for or revenues associated with fossil fuel production, it communicates dissatisfaction among stakeholders which ultimately threatens fossil fuel firms' reputation and legitimacy. The effect of social movements on market prices is even stronger in financial markets with higher social norms. The performance of sin-stocks, for instance, varies considerably between markets with high and low social norms [22,72,73]. Presuming investors are aware of the connection between the fossil fuel industry and climate change, we can expect an underperformance of fossil fuel stocks because of increased investor sentiment, particularly in markets with high social norms [74].

Based on the empirical and theoretical findings presented above, this study examines whether divestment announcements have an impact on the share price of fossil fuel stocks using the event study methodology. If divestment announcements create abnormal returns of fossil fuel shares, the efficient market hypothesis attests that the market perceives the information as material to the industry (alternative hypothesis).

## 3. Method

The event study methodology compares stock prices on a specific event day to their expected returns [75], assuming an efficient market in which prices reflect all available information. The standard stepwise methodology applied in this study closely replicates a framework highlighted by MacKinlay [38]. The method includes identifying a set of comparable exogenous events and endogenous stock samples, calculating expected returns across an estimation window, and measuring the statistical differences between actual returns and expected returns.

### 3.1. Sample and Data

This study analyzes announcements of pledges to divest, endorsements, and events related to divestment campaigns between 1 January, 2012, and 31 December, 2015. Recognizing that events do fall within a period of general underperformance for the industry, we capture a broad collection of events as early as the inception of Bill McKibben's 'Do the Math' campaign until to the symbolic four-year anniversary of the fossil fuel campaign. This period includes windows of both rising and falling fossil fuel share prices. Events with a positive cumulative average abnormal return over the estimation period relative to the market model were uncorrelated with significant abnormal returns on the day of the event (explained in the results below).

As of 2016, the Fossil-Free organization [76] lists 538 pledges, 23 high profile endorsements, and nearly 1000 national and local divestment campaigns. To extract the most important events, inclusionary and exclusionary screening criteria were applied.

A three-step process of inclusionary screening was conducted. First, news articles published in the Wall Street Journal or Financial Times were identified because of their financial relevance [77]. We extracted 42 publications quoting 'fossil fuel divestment' from the LexisNexis database. Next, the Google Trends database was queried for the word 'divestment' to identify instances of rising public discourse as early warning signs of stock market movements. Queries were conducted on a monthly basis (January 2012 to December 2015) to identify specific days in each month where the discourses of fossil fuel divestment were more frequent. The day with the greatest public discourse from each of the 48 months was cross-checked for relevant news articles on the LexisNexis database. Finally, an additional 29 reports of grey literature, such as publications by the Oxford University's Stranded Assets Program, were identified through a standard Google Search, as key drivers in stimulating discourse on fossil-fuel divestment. This process of inclusionary screening identified 119 unique events.

To scope the events, four steps of exclusionary screening were applied. First, publications of general reference (i.e., fossil-fuel divestment is ... ) or publications that did not include new information compared to former events were excluded. Second, if more than one publication referenced the same event, the earliest publication was considered. Third, we excluded announcements of piecewise developments or calls to action (i.e. 'Fossil-fuel divestment discussion moves toward Board'). Finally,

we did not consider rejections of divestment to guarantee that all events send signals to the market that follow the same direction. As a result, the sample consists of thirteen pledges, five endorsements, and six campaigns (Table 1).

**Table 1.** List of divestment-related events analyzed.

| Date | Divestment Pledge |
| --- | --- |
| 2013-05-14 | Kissel, M. (2013). Opinion: Swarthmore Under Student Siege. Wall Street Journal. |
| 2014-05-06 | Crooks, E. (2014). Stanford endowment votes to sell coal mining shares. Financial Times. |
| 2014-07-11 | Vaughan, A. (2014) World Council of Churches rules out fossil fuel investments. The Guardian. |
| 2014-06-25 | Medact. (2014). UK Doctors Vote to End Investments in the Fossil Fuel Industry. |
| 2014-09-22 | Calia, M. (2014). Rockefeller Fund Seeks to Shed Fossil-Fuel Investments. Wall Street Journal. |
| 2014-10-07 | Smyth, J. (2014). Australian pension fund LGS drops coal assets. Financial Times. |
| 2014-10-08 | Brooks, L. (2014) Glasgow becomes the first university in Europe to divest from fossil fuels. The Guardian. |
| 2014-11-23 | Marriage, M. (2014) Norway's largest pension fund vows to drop coal mine holdings. Financial Times. |
| 2015-04-01 | Mance, H & Clark, P. (2015). Guardian Media Group to sell shares linked to fossil fuels. Financial Times. |
| 2015-04-30 | Clark, P. (2015). Church of England blacklists coal and tar sands investments. Financial Times. |
| 2015-05-18 | Clark, P. (2015). University of Oxford to spurn coal and tar sands investments. Financial Times. |
| 2015-06-23 | Lutheran World Federation. (2015). LWF announces the decision not to invest in fossil fuels. LWF. |
| 2015-10-20 | Carrington, D. (2015). Oslo divests from coal companies. The Guardian. |

| Date | Divestment Endorsement |
| --- | --- |
| 2013-05-01 | Klein, N. (2013). Naomi Klein: Time for Big Green to Go Fossil Free. The Nation. |
| 2014-04-10 | Tutu, D. (2014). Desmond Tutu: We need an apartheid-style boycott to save the planet. The Guardian. |
| 2014-08-06 | Gore, A. & Blood, D. (2014). Strong economic case for coal divestment. Financial Times. |
| 2014-11-04 | Ki-moon, B. (2014) Ban Ki-Moon Endorses Fossil Fuel Divestment. UNFCCC. |
| 2015-03-15 | Carrington, D. (2015). Climate change: UN backs fossil fuel divestment campaign. The Guardian. |

| Date | Divestment Campaign |
| --- | --- |
| 2012-09-19 | 350.org (2012). Do the Math: We're jumpstarting a new movement, and we need your help. 350.org. |
| 2013-10-07 | Ansar, A., Caldecott, B., & Tilbury, J. (2013). Stranded assets and the fossil-fuel divestment campaign: what does divestment mean for the valuation of fossil fuel assets? Stranded Asset Program, SSEE, University of Oxford. |
| 2014-08-25 | Bullard, N. (2014). Fossil-fuel divestment: a $5 trillion challenge. Bloomberg New Energy Finance. |
| 2014-09-19 | Foderaro, L.W. (2014). New York Climate March: Taking a Call for Climate Change to the Streets. New York Times. |
| 2015-02-12 | Mathiesen, K; Howard, E; & Shabbir, N. (2015). Global Divestment Day: 'We are ready for urgent action on climate change'. The Guardian. |
| 2015-09-22 | Crooks, E. (2015). Funds worth $2.6tn pledge to dump coal. Financial Times. |

This table lists the divestment-related events selected for analysis from January 2012 to December 2015. Announcements are classified as pledges, endorsements, or campaigns. The earliest date of publication and its associated reference are presented.

### 3.2. Event Windows

Generally, the event study method assumes that new information is unanticipated. The earliest instance of disclosure, however, may be difficult to identify for divestment announcements that might be the result of lengthy public debates. Appropriate event window selection can mitigate issues around clustering and confounding effects related to autocorrelation among shares and between events.

Regarding confounding effects across independent announcements, we checked for other events within a three-day window immediately surrounding the selected events in line with Meznar, Nigh, and Kwok [57]. Moreover, only short-term effects of divestment announcements are assessed, to control for the sample of companies affected by at least one other event over the time window considered. Confounding effects can also be masked by averaging returns across a large sample size [43]. Finally, the estimation window uses the capital asset pricing model [78] over the mean returns model, such that idiosyncrasies from prior events do not influence future events.

Regarding clustering effects arising from a sample of firms from a common industry, a large sample size and short event windows can adequately control for industry effects [43]. Cross-sectional dependence in abnormal returns within the sample are addressed through the BMP statistical test [41] explained below. While the impact of clustering events can also be mitigated by including an industry

variable in the select expected return models [79], the inclusion of an industry variable may depress the excess returns and lead to a false negative type II error.

This study adopts a combination of a single day (0) event windows, multi-day event window spanning ten days around the event, and an estimation windows of 250 trading days (approximately one year) before the event window.

### 3.3. Sample Selection

We used publicly listed companies of Carbon Underground 200 [80] for our analysis. The Carbon Underground 200 is a list of the top 100 coal and top 100 oil and gas companies worldwide, ranked by the carbon emissions content of their reported reserves. The selection is an adequate representation of publicly traded fossil fuel companies since the largest corporations account for the largest share of potential fossil fuel production and greenhouse gas (GHG) emissions [81–83]. Moreover, the Carbon Underground 200 sample of corporations includes 98 percent of all coal and gas reserves and 97 percent of the oil reserves held by listed companies [80]. The sample size of 200 firms is large enough to suppress the idiosyncratic influences of individual firms and for using statistical tests. Finally, large emitters from the fossil fuel industry are in the focus of divestment campaigns.

Stock returns are collected from the Wharton Research Data Services (WRDS) platform, which hosts the Center for Research in Security Prices (CRSP) database of daily stock prices. End of day returns were adjusted for the effects of stock splits, mergers, and dividends. Prices for the MSCI Inc. all-country world index were used as a benchmark since this index most accurately reflects global market performance [84] and because idiosyncrasies from prior events would be independent of future events [78]. Since absolute financial returns cannot be compared directly because of different market sizes and currencies, all outputs were normalized into continuously compounded returns, to best conform to normality assumptions.

### 3.4. Expected and Abnormal Returns

The expected returns model applied in this study is the capital asset pricing model [62]. We used the MSCI Inc. all-country world index and the one-year US treasury price over an estimation window of 250 trading days before the event window as the external market index and as the risk-free return rate respectively. The expected returns model mitigates instances of non-normality by using logarithmic continuous compounding returns. Finally, mean and market ordinary least squares (OLS) models (see Function 1) are applied as robustness checks for instances of serial correlation and nonsynchronous trading [85].

$$R_{it} = a_i + \beta_i R_{mt} + \varepsilon_{it} \tag{1}$$

where the return ($R_{it}$) of an individual stock ($i$) at time ($t$) is a function of a specific risk-free rate ($a_i$), reference market return ($R_{mt}$) on day $t$ with firm-specific regression coefficient ($\beta_i$), and uncorrelated error term ($\varepsilon_{it}$).

The abnormal return (AR) is the difference between the expected and actual return of a given firm on the event day (Function 2). However, this simple calculation only measures abnormal returns for one security at a time, only accounts for one trading day, and does not address distributional errors.

$$AR_{it} = R_{it} - (a_i + \beta_i R_{mt}) \tag{2}$$

where the abnormal return ($AR_{it}$) of an individual stock ($i$) at time ($t$) is a difference between the actual return of an individual stock ($R_{it}$) and the expected return given the specific risk-free rate ($a_i$), reference market return ($R_{mt}$) on day $t$ with firm-specific regression coefficient ($\beta_i$).

In contrast, cross-sectional aggregation combines abnormal returns of multiple firms in a sample into one average abnormal return (AAR) (Function 3). Furthermore, time series aggregation can extend the event window by combining abnormal returns over many days to one cumulative abnormal return (CAR) (Function 5). Therefore, abnormal returns were aggregated to a cumulative average abnormal

return (CAAR) that takes both, time and the number of shares, into account (Function 6). Finally, in line with other studies, we used the standardized abnormal return (SAR) to divide the residuals by their standard error for normalizing the data [86] (Function 4).

$$AAR_{it} = \frac{1}{N} \sum_{i=1}^{N} AR_{it} \tag{3}$$

where the average abnormal return ($AAR_t$) across stocks at time ($t$) is the sum of abnormal returns ($AR_{it}$) of individual stocks ($i$) at time ($t$) divided by the total number of non-missing stocks ($N$) in the sample.

$$SAR_{it} = AR_{it}/SD_{it} \tag{4}$$

where the standardized abnormal return ($SAR_{it}$) of an individual stock ($i$) at time ($t$) is the abnormal return ($AR_{it}$) of individual stock ($i$) at time ($t$) divided by the standard deviation ($SD_{it}$) of the individual stock ($i$) at time ($t$).

$$\text{Where, } SD_{it} = S_i^2 \sqrt{1 + 1/T_i + \frac{(R_{mt} - R_m)^2}{\sum_{t=1}^{T}(R_{mt} - R_m)^2}}$$

where the standard deviation ($SD_{it}$) of the individual stock ($i$) at time ($t$) is a function of the residual variance from the firm ($s_i^2$) adjusted by the forecast error; $R_{mt}$ is the return on the stock market index at time ($t$), $R_m$ is the average return from the market portfolio in the estimation period, and $T$ is the number of days in the estimation period.

$$CAR(t_1, t_2) = \sum_{t=t_1}^{t_2} AR_{i,t} \tag{5}$$

where the cumulative abnormal return ($CAR$) is the sum of the abnormal return ($AR_{it}$) over the event window between the first ($t_1$) and last ($t_2$) day.

$$\text{CAAR} = \frac{1}{n} \sum_{i=1}^{n} CAR(t_1, t_2) \tag{6}$$

where the cumulative average abnormal return ($CAAR$) is the sum of cumulative abnormal returns ($CAR$) divided by the sample size ($n$).

### 3.5. Statistical Model

In line with other event studies, this study adopts a standardized cross-sectional (BMP) test as the primary significance test [41]. Tests of standardized returns have greater statistical power than traditional parametric tests [38,41,87] because of their applicability in instances of cross-sectional correlation and event-induced volatility. Since fossil fuel divestment announcements share common event days and address only one industry, parametric tests will frequently reject the null hypothesis of no effect resulting in a type I error [42]. Standardized parametric tests are also preferred to non-parametric tests when abnormal returns are normalized and continuous. The BMP statistical test [41], applied in this study, uses an ordinary cross-sectional approach to identify event-induced increases in variance [42,43] but replaces the event day cross-sectional standard deviation in the event window with standardized residuals over the estimation window [88]. This method adjusts for event-induced variance realized when announcements share common event days or stocks share a

common industry. Finally, the adjusted BMP test depresses the test statistic when the general market correlates with industries, as in the case of the fossil fuel industry (Function 7).

$$Z_{BMP,t} = \frac{1/n \sum_{i=1}^{n} SAR_{it}}{\sqrt{1/n(n-1) \sum_{i=1}^{n} \left( SAR_{it} - \sum_{i=1}^{n} SAR_{it}/N \right)^2}} \tag{7}$$

where $Z_{BMP,t}$ is the BMP test statistic as calculated by the standardized abnormal return ($SAR_{it}$).

Robustness tests, such as the cross-sectional t-test, the crude dependence test, the Patell test, and the sign test were conducted as robustness checks.

## 4. Results

The following section reports on the results of this study. First, we present the descriptive statistics of the sample set. Then we inform about the results of the independent and aggregate statistical tests.

### 4.1. Descriptive Statistics

Overall, we analyzed 24 independent event announcements between 2012 and 2015 over a sample of 199 coal, oil, and gas firms. Of the 200 publicly traded companies listed by Carbon Underground 200, one stock ticker (coAL) was not accessible on the Wharton Research Data Services (WRDS) database.

Table 2 presents the descriptive statistics of the entire sample and the benchmark during the entire period and the event windows. The mean and median returns are zero, the standard deviation is approximately 0.011, and the distribution of the sample is leptokurtic and slightly skewed to the left. The data suggests that the financial returns of the Carbon Underground 200 firms between 2012 and 2015 are similar to the benchmark. The data across 521 days (pre- and post-event windows) surrounding the event periods, however, suggests that the financial returns of Carbon Underground 200 firms are lower than the benchmark.

**Table 2.** Descriptive statistics of the sample and benchmark between 2012 and 2015, and during the event periods.

| Descriptive Statistics | 'Carbon Underground' between 2012 and 2015 | MSCI ACWI Benchmark between 2012 and 2015 | 'Carbon Underground' Around Events | MSCI ACWI Benchmark Around Events |
|---|---|---|---|---|
| Average Daily Return of Security | 0.000 | 0.000 | −0.011 | 0.017 |
| Cumulative Return Over Time | −0.534 | 0.035 | −0.293 | 0.461 |
| Median | 0.000 | 0.000 | −0.005 | 0.019 |
| Standard Deviation | 0.012 | 0.005 | 0.062 | 0.051 |
| Skewness | −0.234 | −0.986 | −0.183 | −0.326 |
| Kurtosis | 9.765 | 9.695 | −0.643 | −0.238 |

This table reports summary statistics of daily returns for the Carbon Underground 200 fossil fuel stocks sampled in this study and the MSCI All-Country World Index benchmark. The summary statistics are the averages of estimates (mean, cumulative return, median, standard deviation, skewness, and kurtosis). The table reports summary statistics for the breadth of the entire dataset from January 2012 to December 2015 as well as aggregated summary statistics of event windows spanning -260 to 260 days around event days.

Not all events underperformed relative to the market index. Seven of the twenty-four events had a positive cumulative average abnormal return (CAAR) over the estimation period (-250, -11) relative to the market model. We conducted a t-test to compare whether events during a phase of increasing return over the estimation window had a statistically significant different return on the day of the event relative to events during a phase of decreasing returns (p = 0.28). The result demonstrates that negative abnormal returns on the event day are independent of whether the sample of stocks underperformed across the estimation window. Further analysis of aggregated abnormal returns between positive and negative estimation windows is conducted below. Many events did not notice a statistically significant negative drop; eleven events outperformed the expected return on the event day (Figure 1).

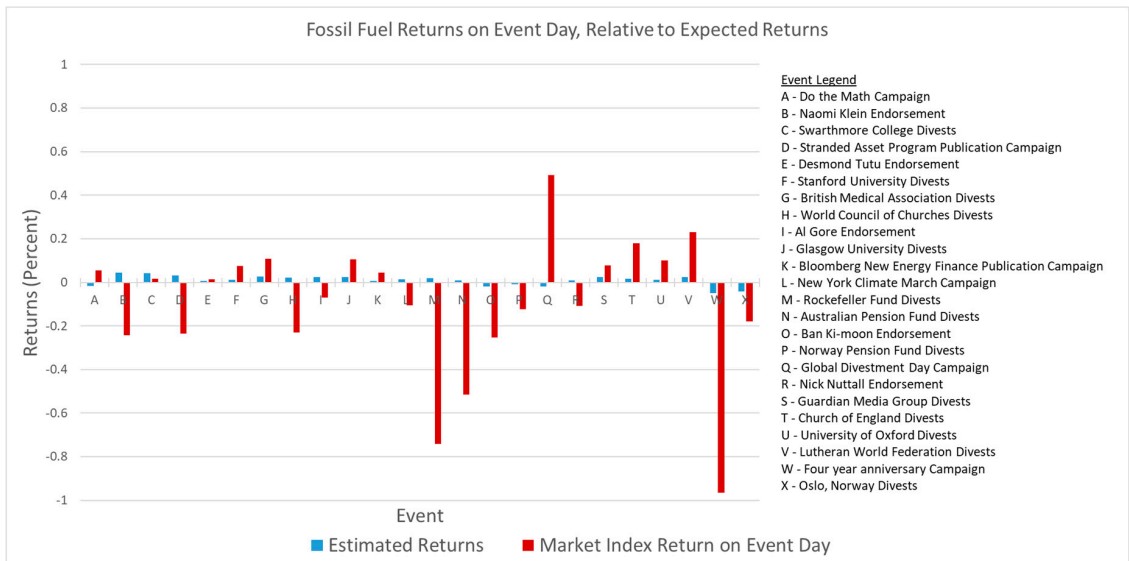

**Figure 1.** Fossil fuel returns on the event day relative to expected returns. This graph shows the cumulative average abnormal returns of fossil fuel stocks on the event day relative to the expected returns calculated over the estimation window.

The following tests measure the statistical significance of the difference between abnormal returns surrounding divestment events. The results suggest significant lower returns of fossil fuel shares compared to the benchmark.

*4.2. Independent Tests*

Independent tests examine the abnormal returns of each event individually. Cumulative abnormal returns (CAR) are measured across one and ten days around the event date, using the capital asset pricing model and standardized cross-sectional BMP tests. One-day intervals capture causal effect while ten-day intervals capture longer-term perceptions. Table 3 indicates that many but not all divestment events have a significant effect on share prices.

The first observation from the individual results is that some events are more influential than others. Of the selected events, the most influential one fell around the third weekend of September 2014, which included the New York People's Climate March, the Rockefeller Fund's divestment announcement, and the 2014 UN climate summit. In contrast, the least influential events include the Global Divestment Day campaign and the Guardian Media Group's divestment announcement.

Furthermore, there is a tipping point at which divestment events have longer-term impacts on fossil fuel stocks. Events after 19 September 2014 (the New York Climate March) have longer-term negative abnormal returns spanning as far as ten days around the event day, indicating a shift in the investors' long-term perceptions of divestment events over the last year. Additional analyses on abnormal returns between events prior to and post the New York Climate March are conducted below.

**Table 3.** One-day and ten-day effects of divestment events on fossil fuel share price by event.

| Date | Announcement | (−1,1) | | | (−10,10) | | |
|---|---|---|---|---|---|---|---|
| | | **CAAR** | **BMP** | *p*-Value | **CAAR** | **BMP** | *p*-Value |
| 2012-09-19 | Do the Math Campaign | −0.008 | 7.54 | <0.001 ** | 0.017 | 6.73 | 1.000 |
| 2013-05-01 | Naomi Klein Endorsement | −0.003 | 2.91 | 0.002 ** | 0.008 | 2.66 | 0.996 |
| 2013-05-14 | Swarthmore College Divests | −0.005 | 3.37 | <0.001 ** | 0.006 | 1.90 | 0.972 |
| 2013-10-07 | SAP Publication Campaign | −0.003 | 2.59 | 0.005 ** | −0.002 | 0.81 | 0.208 |
| 2014-04-10 | Desmond Tutu Endorsement | 0.000 | 0.31 | 0.620 | 0.009 | 2.66 | 0.996 |
| 2014-05-06 | Stanford University Divests | 0.001 | 0.61 | 0.729 | −0.005 | 1.33 | 0.092 |
| 2014-07-11 | World Council of Churches Divests | −0.002 | 1.67 | 0.047 ** | −0.003 | 0.85 | 0.197 |
| 2014-06-25 | British Medical Association Divests | −0.002 | 2.15 | 0.016 ** | −0.001 | 0.42 | 0.337 |
| 2014-08-06 | Al Gore Endorsement | −0.005 | 2.42 | 0.008 ** | 0.005 | 1.62 | 0.947 |
| 2014-08-08 | Glasgow University Divests | 0.002 | 1.68 | 0.953 | −0.003 | 1.10 | 0.137 |
| 2014-08-25 | BNEF Publication Campaign | −0.001 | 1.19 | 0.117 | 0.004 | 0.92 | 0.821 |
| 2014-09-19 | New York Climate March Campaign | −0.010 | 5.99 | <0.001 ** | −0.042 | 8.47 | <0.001 ** |
| 2014-09-22 | Rockefeller Fund Divests | −0.011 | 6.34 | <0.001 ** | −0.046 | 9.57 | <0.001 ** |
| 2014-10-07 | Australian Pension Fund Divests | −0.005 | 4.95 | <0.001 ** | −0.043 | 11.66 | <0.001 ** |
| 2014-11-03 | Ban Ki-moon Endorsement | −0.001 | 0.62 | 0.269 | 0.002 | 0.50 | 0.691 |
| 2014-11-24 | Norway Pension Fund Divests | 0.001 | 0.67 | 0.749 | −0.057 | 8.22 | <0.001 ** |
| 2015-02-12 | Global Divestment Day Campaign | 0.009 | 7.81 | 1.000 | 0.025 | 4.27 | 1.000 |
| 2015-03-16 | Nick Nuttall Endorsement | −0.006 | 3.29 | 0.001 ** | −0.025 | 7.38 | <0.001 ** |
| 2015-04-01 | Guardian Media Group Divests | 0.003 | 1.56 | 0.941 | 0.036 | 9.29 | 1.000 |
| 2015-04-30 | Church of England Divests | 0.003 | 1.79 | 0.963 | −0.011 | 2.54 | 0.006 ** |
| 2015-05-18 | University of Oxford Divests | −0.006 | 3.57 | <0.001 ** | −0.022 | 5.07 | <0.001 ** |
| 2015-06-23 | Lutheran World Federation Divests | 0.005 | 2.95 | 0.998 | −0.059 | 8.82 | <0.001 ** |
| 2015-09-22 | Four-year anniversary Campaign | −0.019 | 8.56 | <0.001 ** | 0.004 | 1.03 | 0.849 |
| 2015-10-20 | Oslo, Norway Divests | −0.015 | 7.02 | <0.001 ** | 0.015 | 3.65 | 1.000 |

This table presents the cumulative average abnormal returns (CAAR) for each independent event, relative to the MSCI All-Country World Index market benchmark. The CAAR aggregates results over the sample of 200 fossil fuel stocks and across two different time periods (one-day and ten-day intervals respectively). The fourth and seventh columns present the adjusted standardized cross-sectional (BMP) test, which is used as the primary significance test in this study (Boehmer et al., 1991). The fifth and eighth columns provide the associated *p*-value. * and ** denote significance at the 5 and 1% level, respectively.

## 4.3. Aggregate Tests

Aggregate tests cluster the returns of divestment events around their respective event dates. A statistically significant abnormal return across an aggregate of events indicates that its influence is significant.

Abnormal returns are assessed up to 10 days before and after the event date, day 0. Using the capital asset pricing model and standardized cross-sectional BMP tests, the results indicate an average abnormal return of −0.127 percent on the event day 0. This result is statistically significant ($p = 0.040$). Table 4 presents the distribution of daily abnormal returns around divestment events. The statistically significant negative abnormal return indicates a general effect across events. Confounding effects are masked by averaging the returns of the sample. Furthermore, the negative cumulative average abnormal returns one day before and after the event are statistically significant ($p = 0.026$, BMP = 1.95); this captures any lag effects of the announcement. The negative cumulative average abnormal return ten days before and after the event is not statistically significant, suggesting that the overall impact of divestment events are short-term.

Figure 2 presents the average performance of fossil fuel shares, normalized across 24 independent event dates, relative to the MSCI market benchmark. The figure suggests that fossil fuel returns are depressed during the event and post-event window.

Aggregate results were further examined to identify variation between events with a positive and negative estimation window (Table 5). Over the one-day interval, events with both positive and negative estimation windows realized a statistically significant negative cumulative average abnormal return ($p = 0.017$ and $p = 0.032$ respectively). Events with positive estimation windows relative to the benchmark also realized a statistically significant negative cumulative average abnormal return over the ten-day interval ($p = 0.016$), whereas the one-day interval did not. By controlling for

positive estimation windows, we are confident that event impacts are independent of the general underperformance of the industry over the time-period analyzed.

**Table 4.** Abnormal returns (AR) and cumulative abnormal returns (CAR).

| Event Day/Interval | AAR | CAAR | BMP Test | *p*-Value |
|---|---|---|---|---|
| −260 | −0.025 | −0.025 | | |
| −10 | 0.066 | 0.041 | 0.537 | 0.296 |
| −9 | 0.045 | 0.086 | 0.959 | 0.169 |
| −8 | 0.049 | 0.126 | 0.239 | 0.406 |
| −7 | −0.038 | 0.087 | −0.427 | 0.335 |
| −6 | −0.109 | −0.021 | 1.084 | 0.140 |
| −5 | −0.037 | −0.059 | 0.373 | 0.355 |
| −4 | 0.044 | −0.016 | 0.774 | 0.220 |
| −3 | −0.022 | −0.038 | 1.085 | 0.140 |
| −2 | −0.095 | −0.133 | 0.244 | 0.404 |
| −1 | −0.054 | −0.187 | 1.253 | 0.106 |
| 0 | −0.127 | −0.314 | 1.758 | 0.040 * |
| 1 | −0.148 | −0.462 | 1.221 | 0.112 |
| 2 | −0.026 | −0.488 | 0.091 | 0.464 |
| 3 | 0.004 | −0.492 | 0.324 | 0.373 |
| 4 | −0.177 | −0.669 | 0.462 | 0.322 |
| 5 | −0.098 | −0.767 | −0.325 | 0.373 |
| 6 | 0.026 | −0.741 | 0.609 | 0.272 |
| 7 | −0.013 | −0.755 | 0.127 | 0.450 |
| 8 | −0.083 | −0.837 | 1.163 | 0.123 |
| 9 | 0.029 | −0.808 | 0.609 | 0.271 |
| 10 | −0.006 | −0.814 | 0.196 | 0.422 |
| | | | | |
| (−1,1) | | −0.327 | 1.950 | 0.026 * |
| (−10,10) | | −0.792 | 1.293 | 0.099 |

This table presents the aggregated average abnormal returns (AAR) and cumulative average abnormal return (CAAR) across all events, relative to the MSCI All-Country World Index market benchmark. The adjusted standardized cross-sectional (BMP) test is used as the primary significance test in this study (Boehmer et al., 1991). The associated *p*-value is presented. * and ** denote significance at the 5 and 1% level, respectively.

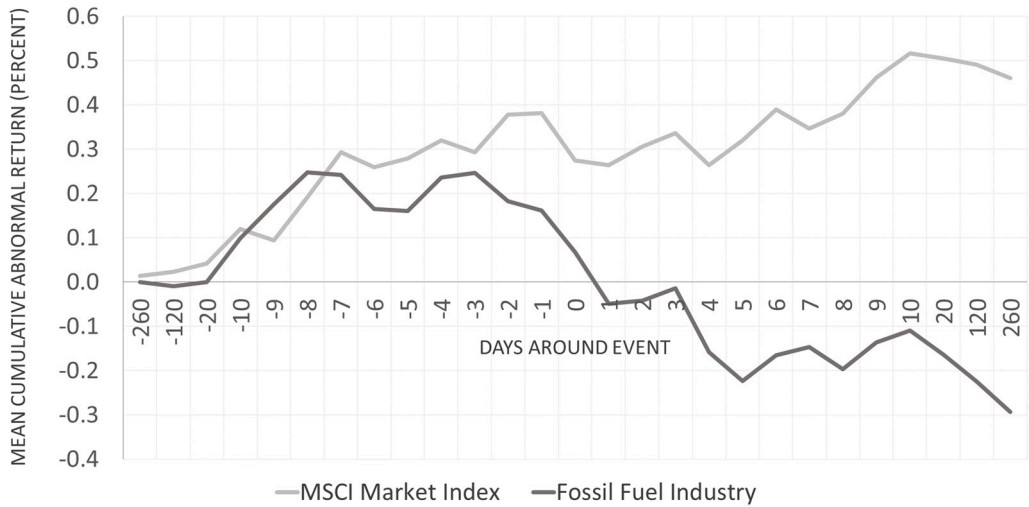

**Figure 2.** Effect of Divestment Events on Share Price. This graph shows the aggregated cumulative average abnormal return of the share price of fossil fuel stocks relative to the MSCI All-Country World Index, from the beginning of the estimation window -260 days prior to the event day to 260 days after the event day. Day 0 indicates the event day.

Aggregate results were also examined to identify variation between types of events (Table 5). Campaigns, pledges, and endorsements, all realized statistically significant negative cumulative average abnormal returns over the one-day interval ($p = 0.001$, $p = 0.014$, and $p = 0.038$ respectively). However, these results did not hold over the ten-day interval.

Finally, there is a notable variation between events prior to and post the New York Climate March (Table 5). While events realized statistically significant negative cumulative average abnormal returns for both categories over the one-day interval ($p = 0.027$, $p = 0.025$ respectively), only post-events realized statistically significant negative cumulative average abnormal returns for both categories over the ten-day interval ($p < 0.000$). Thus, events after the New York Climate March maintain longer-term effects, signaling a shift in the influence of divestment announcements made as well as investors' long-term perception of divestment announcements.

**Table 5.** One-day and multi-day effects of divestment events on fossil fuel share price by event.

| | (−1,1) | | | (−10,10) | | |
|---|---|---|---|---|---|---|
| | **CAAR** | **BMP** | ***p*-Value** | **CAAR** | **BMP** | ***p*-Value** |
| **PANEL A** | | | | | | |
| All Events | −0.327 | 1.950 | 0.026 * | −0.792 | 1.293 | 0.099 |
| **PANEL B** | | | | | | |
| Events with Positive Estimation Windows | −0.393 | 2.146 | 0.017 ** | −1.230 | 2.149 | 0.016 ** |
| Events with Negative Estimation Windows | −0.300 | 1.869 | 0.032 ** | −0.611 | −0.940 | 0.174 |
| **PANEL C** | | | | | | |
| Campaign Events | −0.534 | 3.010 | 0.001 ** | 0.082 | 0.610 | 0.271 |
| Pledge Events | −0.344 | 2.218 | 0.014 * | −0.688 | 0.901 | 0.185 |
| Endorsement Events | −0.283 | 1.784 | 0.038 * | 0.035 | 0.011 | 0.496 |
| **PANEL D** | | | | | | |
| Events prior to NY Climate March | −0.250 | 1.931 | 0.027 ** | 0.300 | 1.088 | 0.139 |
| Events post NY Climate March | −0.392 | 1.966 | 0.025 ** | -1.716 | 3.307 | 0.000 ** |

This table presents the aggregated cumulative average abnormal return (CAAR) across all events, relative to the MSCI All-Country World Index market benchmark. The CAAR aggregates all events over the sample of 200 fossil fuel stocks and across two different time periods (one-day and ten-day intervals, respectively). The third and sixth columns present the adjusted standardized cross-sectional (BMP) test, which is used as the primary significance test in this study (Boehmer et al., 1991). The fourth and seventh columns provide the associated *p*-value. * and ** denote significance at the 5 and 1% level, respectively. Panel A categorizes all events together over the one-day and ten-day intervals. Panel B categorizes events based on whether fossil fuel stocks had a positive estimation window relative to MSCI Inc. All-Country World Index market benchmark, to differentiate between periods of underperformance. Panel C categorizes events based on their announcement type, to identify variation in influence. Panel D categorizes events by time-period, differentiating between events prior to and post the New York Climate March.

We note that the results used the capital asset pricing model for the estimation window and the BMP test as the statistical model. However, the analysis was also conducted using the mean returns and market OLS estimation models as well as the cross-sectional t-test, crude dependence test, Patell test, and sign test as statistical tests. In all, 15 robustness checks were conducted; we present the results of these checks in Table 6 below. Discrepancies in results are due to the precision of various estimation models and statistical tests as explained later. The mean returns model is exposed to idiosyncrasies among fossil fuel shares, thus depressing the excess returns and resulting in false negative test results. The market model and Capital Asset Pricing Model (CAPM) yield similar results; however, the latter is slightly more sensitive as it accounts for market risks. Tests of standardized returns (including the crude dependence test, Patell test, and BMP test) are preferred for their ability to account for cross-sectional correlation and event-induced variance, which is important when the sample shares common event days or a common industry.

**Table 6.** Robustness tests by expected return models and statistical models on the day of the announcement aggregated across all events.

| Expected Returns Models | Statistical Test | t-Value |
|---|---|---|
| Mean Model | CSect | −0.18 |
| | CDA | −0.47 |
| | PatellZ | −0.13 |
| | BMPZ | 0.06 |
| | Sign | 0.585 |
| Market OLS | CSect | −1.11 |
| | CDA | −4.58 ** |
| | PatellZ | −1.65 * |
| | BMPZ | −1.51 |
| | Sign | −0.55 |
| Capital Asset Pricing Model | CSect | −1.43 |
| | CDA | −5.73 ** |
| | PatellZ | −1.88 * |
| | BMPZ | −1.76 * |
| | Sign | −1.476 |

This table presents the results of alternative expected returns models and statistical tests on the event day. Primarily, this study adopts the Capital Asset Pricing Model and the adjusted standardized cross-sectional (BMP) test (Boehmer et al., 1991). Alternate combinations result in conflicting results because of their applicability in instances of cross-sectional correlation and event-induced volatility. The fourth and seventh columns provide the associated *p*-value. * and ** denote significance at the 5 and 1% level, respectively.

## 5. Discussion

This study contributes to the knowledge on the impact of ethical investment behavior on share prices of fossil fuel stocks and consequently on their ability to explore and exploit fossil fuel resources. Thus, it addresses a gap that is neglected in conventional finance research [89]. Our results suggest that prominent divestment announcements have a statistically significant negative impact on the price of fossil fuel shares. Together with the findings by Ekwurzel et al. [81] who trace nearly two-thirds of total industrial carbon dioxide and methane emissions to 90 major industrial carbon producers, divestment has the potential to influence supply-side emissions of the major climate change contributors.

The results of this study make several contributions to the study of fossil fuel divestment. First, divestment announcements can have both short-term impacts (through one-day intervals that capture causal effect) as well as longer-term impacts (through ten-day intervals that capture longer-term perceptions). Longer-term impacts are more prevalent in later events, suggesting a shift in investor perception as divestment announcements gained legitimacy. All types of divestment announcements analyzed were found to impact share prices. Finally, abnormal returns are independent of general underperformance of the industry (as tested in the aggregate results), as well as clustering and confounding effects (moderated through the BMP statistic).

Our results are in line with the studies by Wright and Ferris [60] and Meznar, Nigh, and Kwok [90] that announcements of divestment lead to negative returns. Thus, the findings of this study complement the results of significant negative returns in the short term but also suggests a long-term impact.

This study suggests that if powerful and legitimate stakeholders announce divestment because of an urgent matter, such as climate change, they can influence the share prices of affected industries. Similar to announcements about environmental fines [53], or about breaches of codes of conduct [51], divestment announcements affect fossil fuel share prices.

Concerning the social movement theory, media influence may have also played a role in legitimizing secondary stakeholders. Social norms in financial market reinforce this effect [74]. Not only has the divestment movement communicated social norms with regard to financial risks of climate change (such as stranded assets) but so have financial market players, such as the governor of the Bank of England [91] as well as the Task Force on Climate-Related Disclosures [92] who have warned

about climate-related financial risks. Consequently, divestors can contribute to a decline in fossil fuel production that is needed to achieve their climate goals [93].

Of course, the most common criticisms claim that the lower share prices might be caused by market forces or declining oil prices. However, our results find that aggregate abnormal returns are similar between events where fossil fuel stocks under- and outperformed the market index over the estimation windows.

## 6. Conclusions

In line with Harris and Gurel [16], we conclude that divestors can create a demand shift for fossil fuel shares through increasing financial capital costs and decreasing the solvency of fossil fuel firms. Both capital cost and solvency have important impacts on the ability of fossil fuel companies to explore and exploit their resources. The question, however, of whether the decrease in share price influences the business strategy of affected companies is still open and should be addressed in future research. Recent evidence suggests that the divestment movement may not have influenced the decision of Germany's major electric companies to remove select fossil fuel operations from their company portfolio, however, there is a gap in the literature on its effect on fossil fuel companies [94]. Though lower share prices increase the costs of financial capital [95], the effect on the exploration of new fossil fuel resources [34] is still unclear. Furthermore, the impact of divestment on fossil fuel extraction in different parts of the world should be analyzed to address equity considerations regarding fossil fuel extraction [96].

To shareholders, divestment does have a place in the ethical investors' toolbox, as a means to influence the corporate objectives of the fossil fuel industry. However, not all pledges to divest will be equally impactful. For instance, the Guardian Media Group divested its one-billion-dollar fund with little influence, whereas the University of Oxford pledged to divest an endowment which held no shares with much greater impact.

To the advocates of divestment, the results suggest that divestment does have an impact on the industry but does not infer that divestment will have an impact on corporate objectives. To the fossil fuel industry, the fossil-fuel divestment campaign has affected the share price of fossil fuel stocks, as markets perceive these threats as credible to the industry's expected returns. Hence, it is in the industry's best interest to engage with shareholders and stakeholders alike, to address their concerns. To stakeholders, these results encourage continued discourse on the topics of stranded assets and the carbon budget, as equally influential to the divestment movement.

As a strategy, shareholders can pressure the fossil fuel industry by announcing divestment if other strategies, such as shareholder engagement, do not lead to a change in the addressed fossil-fuel firms. Further, to increase the influence of a fossil-fuel divestment announcement, a socially responsible shareholder might establish shareholder associations that announce divestment instead of announcing it individually.

Stakeholders can also pressure the financial industry to offer more fossil-free products, such as mutual funds, to provide fossil-free divestment opportunities. Furthermore, stakeholders might inform conventional investors about the financial risks of being invested in fossil fuels to achieve change.

Generally, it seems that strategies that create awareness of financial market participants are most helpful in markets with high social norms. Instead of 'preaching to the converted' (environmentalists, socially responsible investors), divestment announcements should address financial market participants.

Regarding the divestment literature, this study provides the empirical basis against theoretical literature and provides the groundwork for more detailed empirical analysis in future studies. To reiterate, however, this study cannot make any inference to the long-term effect (10+ days) of divestment due to confounding effects. As such, the results do not confirm that divestment can 'force change' on the industry but that divestment has contributed to depressing share prices of fossil fuel firms.

Numerous opportunities for further research on the topic of divestment can be pursued. First, this study can be extended to more succinctly measure the impact of different events, by further categorizing and comparing between events and subsamples. Moreover, we concede that further analysis of more recent events that were not under a period of downturn should be a priority in this research agenda. Third, the impact of divestment can be compared to the impact of other climate change strategies, such as engagement initiatives. Fourth, even if divestment announcements influence stock prices, this study does not guarantee to meet the climate goals or that the industry has faced financial pressures to reduce emissions.

Similarly, it does not include an analysis of whether divestment is the right move for shareholders. Divestment can also be studied through the lens of the divesting intuition, applying signaling theory to understand why divesting institutions pledge to divest from fossil fuels, even if the action may not be in the divesting firm's favor. Finally, studies could analyze the role of investment and divestment for other climate-related fields, such as land use [97] low carbon feed source alternatives [98], and nutrient availability [99].

**Author Contributions:** The authors contributed to the paper as following "conceptualization, O.W. and T.D.; methodology, O.W. and T.D.; validation, O.W. and T.D.; formal analysis, T.D.; data curation, T.D.; writing—original draft preparation, T.D.; writing—review and editing, O.W. and T.D.; visualization, T.D.; supervision O.W.

**Funding:** This research received no external funding.

**Conflicts of Interest:** The authors declare no conflict of interest.

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
