# Peer review of "The Impact of Divestment Announcements on the Share Price of Fossil Fuel Stocks"

_sustainability, doi:10.3390/su11113122_

Round 1

Reviewer 1 Report

- Please enlarge the abstract - much more could be written here.

- The manuscript lacks of a number of abbreviation explanation, which makes reading not so easy:

  e.g. lines 36,52, 53, 116, 136, 254, 266, 281, 284, 329, 450, 455, and 534

- Figure 1 is not readable

- Please split in two chapter 5. Discussion and conclusions (5. Discussion and 6. Conclusions) and please bring in a reference/s for each statement

Minor issues:

- Line 14: please specify if the 200 companies operate worldwide or are limited to a certain territory  (also at line 262)

- Line 136: do you mean "Similar works..."?

- Please indicate the link at line 212 as reference

- Please bring the caption of Table 3 to the next page together with the table

- Line 472: please write correctly the chemical indications

- Please format correctly the References section

Author Response

Dear reviewer,

Thank you very much for your comments. As per your recommendations, we have made the following changes. The line numbers correspond with the tracked changes document.

Please enlarge the abstract - much more could be written here.

The abstract has been extended by adding more detail about the results of the study (line 16-25): “Events are analyzed independently and in aggregate. The results make several notable contributions. While many events experienced short term negative abnormal returns around the event day, the effects of events were more pronounced over longer event windows following the New York Climate March, suggesting a shift in investor perception. The results also find that divestment announcements related to campaigns, pledges, and endorsements all have a significant effect over the short-term event window. Finally, the results control for the general underperformance of the industry over the estimation window, attesting that the price change is caused by divestment announcements. Several robustness tests using alternate expected returns models and statistical tests were conducted to ensure the accuracy of the result.”

The manuscript lacks of a number of abbreviation explanation, which makes reading not so easy: e.g. lines 36,52, 53, 116, 136, 254, 266, 281, 284, 329, 450, 455, and 534

We have corrected for the undefined abbreviations across the manuscript as follows.

Line 36 - Toronto Stock Exchange (TSX) 260

Line 62, 299 - MSCI Inc.

Line 63 - Boehmer, Musumeci and Poulsen (BMP)

Line 132 - chief executive (CEO)

Line 152 - 21st Conference of Parties (COP21)

Line 283 - greenhouse gas (GHG)

Line 303 - ordinary least squares (OLS)

Line 368 - Wharton Research Data Services (WRDS)

Line 508 - Capital Asset Pricing Model (CAPM)

Line 592 - socially responsible (SRI)

Figure 1 is not readable

We amended figure 1 with a new legend of events along the x-axis, for easier readability.

Please split in two chapter 5. Discussion and conclusions (5. Discussion and 6. Conclusions) and please bring in a reference/s for each statement

Chapter 5 has now been split into discussion and conclusions, and the citations format in lines 524 and 526 is corrected.

Line 14: please specify if the 200 companies operate worldwide or are limited to a certain territory  (also at line 262)

The carbon underground 200 companies operate worldwide and this has been clarified in the manuscript (line 280).

Line 136: do you mean "Similar works..."?

This sentence has been reworded with “similar works” (line 151)

Please indicate the link at line 212 as reference

This link has been converted to a reference (line 228)

Please bring the caption of Table 3 to the next page together with the table

All tables, figures, and captions have been corrected to ensure that they are on the same page in their entirety.

Line 472: please write correctly the chemical indications

The chemical indicators for carbon dioxide and methane (line 526) have been replaced with the text.

Please format correctly the References section

The duplication of numbers in the references section has been corrected.

Thank you again for your time. We hope the revisions are sufficient.

The authors.

Reviewer 2 Report

Paper ˝The impact of divestment announcements on the share price of fossil fuel stocks˝ presents a very interesting study about the share price of fossil fuel stocks. The study is well organized, research methodology is adequate and I can suggest just a few minor changes:

1. Please define every abbreviation first time when you mentioned it in the paper.

2. When you introduce equation in the manuscript please define all the terms in the equations.

3. In the Discussion and Conclusion section, you cite some of your reference with authors names and year of publication instead of literature number.

4. You have the mistake in the reference section. The reference numbers are doubled.

Author Response

Dear reviewer,

Thank you very much for your comments. As per your recommendations, we have made the following changes. The line numbers correspond with the tracked changes document.

Please define every abbreviation first time when you mentioned it in the paper.

We have corrected for the undefined abbreviations across the manuscript as follows.

Line 36 - Toronto Stock Exchange (TSX) 260

Line 62, 299 - MSCI Inc.

Line 63 - Boehmer, Musumeci and Poulsen (BMP)

Line 132 - chief executive (CEO)

Line 152 - 21st Conference of Parties (COP21)

Line 283 - greenhouse gas (GHG)

Line 303 - ordinary least squares (OLS)

Line 368 - Wharton Research Data Services (WRDS)

Line 508 - Capital Asset Pricing Model (CAPM)

Line 592 - socially responsible (SRI)

When you introduce equation in the manuscript please define all the terms in the equations.

We have added descriptors to each of the equations across the manuscript as follows.

Line 306 - Where the return (Rit) of an individual stock (i) at time (t) is a function of a specific risk-free rate (ai), reference market return (Rmt) on day t with firm-specific regression coefficient (βi), and uncorrelated error term (εit).

Line 314 - Where the abnormal return (ARit) of an individual stock (i) at time (t) is a difference between the actual return of an individual stock (Rit) and the expected return given the specific risk-free rate (ai), reference market return (Rmt) on day t with firm-specific regression coefficient (βi).

Line 325 - Where the average abnormal return (AARt) across stocks at time (t) is the sum of abnormal returns (ARit) of individual stocks (i) at time (t) divided by the total number of non-missing stocks (N) in the sample.

Line 329 - Where the standardized abnormal return (SARit) of an individual stock (i) at time (t) is the abnormal return (ARit) of individual stock (i) at time (t) divided by the standard deviation (SDit) of the individual stock (i) at time (t).

Line 333 - Where the standard deviation (SDit) of the individual stock (i) at time (t) is a function of the residual variance from the firm (si2) adjusted by the forecast error.  - wWhere Rmt is the return on the stock market index at time (t), Rm is the average return from the market portfolio in the estimation period, and T is the numbers of days in the estimation period.

Line 338 - Where the cumulative abnormal return (CAR) is the sum of the abnormal return (ARit) over the event window between the first (t1) and last (t2) day.

Line 341 - Where the cumulative average abnormal return (CAAR) is the sum of cumulative abnormal returns (CAR) divided by the sample size (n).

Line 358 - Where ZBMP,t is the BMP test statistic as calculated by the standardized abnormal return (SARit).

In the Discussion and Conclusion section, you cite some of your reference with authors names and year of publication instead of literature number.

The citations format for the references on lines 524 and 526 is corrected.

You have the mistake in the reference section. The reference numbers are doubled

The duplication of numbers in the references section has been corrected.

Thank you again for your time. We hope the revisions are sufficient.

The authors.

Reviewer 3 Report

First of all, I greatly appreciate the initiative of Mr Olaf Weber to launch a special issue with a new theme, namely Divestment and Sustainability.

Regarding the article submitted for review, I think it is very well documented and the conclusions are very well founded. In addition, the article fulfills the conditions of originality and novelty of scientific research.

 I think the text still needs to be arranged so that table 2 and table 3 are better presented on the page.

The bibliographic reference section is very extensive, but nevertheless I recommend that the authors also refer to certain articles published in the MDPI journals as a measure to promote the results obtained by our colleagues.

1. Kiyar D, Wittneben B. Carbon as Investment Risk—The Influence of Fossil Fuel Divestment on Decision Making at Germany’s Main Power Providers. Energies. 2015 Sep;8(9):9620-39.

2. Finley-Brook M, Holloman E. Empowering energy justice. International journal of environmental research and public health. 2016 Sep 21;13(9):926.

3. Butler CD. Philanthrocapitalism: Promoting Global Health but Failing Planetary Health. Challenges. 2019 Jun;10(1):24.

Author Response

Dear reviewer,

Thank you very much for your comments. As per your recommendations, we have made the following changes. The line numbers correspond with the tracked changes document.

I think the text still needs to be arranged so that table 2 and table 3 are better presented on the page.

Tables and captions have been corrected to ensure that they are on the same page in their entirety.

The bibliographic reference section is very extensive, but nevertheless I recommend that the authors also refer to certain articles published in the MDPI journals as a measure to promote the results obtained by our colleagues.

1. Kiyar D, Wittneben B. Carbon as Investment Risk—The Influence of Fossil Fuel Divestment on Decision Making at Germany’s Main Power Providers. Energies. 2015 Sep;8(9):9620-39.

2. Finley-Brook M, Holloman E. Empowering energy justice. International journal of environmental research and public health. 2016 Sep 21;13(9):926.

3. Butler CD. Philanthrocapitalism: Promoting Global Health but Failing Planetary Health. Challenges. 2019 Jun;10(1):24.

We have included the recommended references from the MDPI journals as follows.

Line 110 - Notably, while divestment communicates a symbolic message, concerns have been raised around environmental justice related to decision-making power, rights, and diversity.

Line 112 - Moreover, institutional divestors may be motivated by market-driven solutions than proactively mitigating the financial risks associated with carbon emissions and climate change.

Line 562 - Recent evidence suggests that the divestment movement may not have influenced the decision of Germany’s major electric companies to remove select fossil fuel operations from their company portfolio, however, there is a gap in the literature on its effect on fossil fuel companies.

Thank you again for your time. We hope the revisions are sufficient.

The authors.

Reviewer 4 Report

In the manuscript, authors’ measures abnormal deviations in stock prices of the top 200 oil, gas, and coal companies by proven reserves, on days of prominent divestment announcements, adopting an event study methodology.

The topic of the paper is very interesting as well as the academic contribution of the work.

The paper is clear, well written and well organized. The introduction provide a sufficient background. The methodological approach is technically correct. All the relevant references have been included demonstrating a good knowledge of the field by authors', and the literature review contributes to the definition of the concrete contribution of the paper. The conclusions are supported by the results. English language and style are fine.

Author Response

Reviewer Comments

In the manuscript, authors’ measures abnormal deviations in stock prices of the top 200 oil, gas, and coal companies by proven reserves, on days of prominent divestment announcements, adopting an event study methodology.

The topic of the paper is very interesting as well as the academic contribution of the work.

The paper is clear, well written and well organized. The introduction provide a sufficient background. The methodological approach is technically correct. All the relevant references have been included demonstrating a good knowledge of the field by authors', and the literature review contributes to the definition of the concrete contribution of the paper. The conclusions are supported by the results. English language and style are fine.

Dear reviewer,

Thank you very much for your comments. We have made some stylistic changes as recommended by other reviewers. We hope the revisions are sufficient.

- The authors.

Round 2

Reviewer 1 Report

The melioration indications have been well implemented.